# The effect of framing on attitudes towards Alzheimer's disease. A comparative study between younger and older adults

**Fátima Cuadrado**[1,2]* , **Adoración Antolí**[1,2], **Bernardino Fernández-Calvo**[1,2]

**1** Department of Psychology, University of Cordoba, Cordoba, Spain, **2** Maimonides Biomedical Research Institute of Cordoba (IMBIC), Cordoba, Spain

☯ These authors contributed equally to this work.

* fatima.cuadrado@uco.es

## Abstract

The stigma and negative attitudes surrounding Alzheimer's disease (AD) are reinforced by the prevalence of their negative representations. This study aimed to determine how AD framing influences attitudes towards AD and whether this influence differs between younger and older people. Additionally, the elaboration likelihood model (ELM) was used to examine the mediating role that emotions induced by different frames may have in bringing about attitude change. Posters with framed messages on AD (dualism and unity) were designed and shown to 136 participants (68 younger and 68 older adults). Both the younger and older participants were randomly divided into two sub-groups. Each sub-group was shown posters of a campaign with different AD frames: one group viewed posters with messages of the dualism frame and the other group viewed posters with messages of the unity frame. To analyse the effect of the different frames, a mixed design of repeated measures (ANOVA) was used in which attitudes towards AD were measured on two occasions. Both the impact and the emotions produced by exposure to the messages were recorded after the presentation of the posters and a MANOVA test was performed on them. Attitudes, impact and emotions experienced by the younger and older participants were compared. Older adults displayed positive attitudes towards AD but less than younger people. Unity-framed messages produced a positive change in attitudes regardless of the audience's age and led to higher levels of happiness, whereas dualism-framed messages had a greater impact and produced feelings of sadness, anger and fear but did not change attitudes. These findings suggest that reframing of AD may be essential to achieve a positive attitudinal change in both younger and older populations and foster positive emotions. The use of unity-framed messages should be considered when developing and implementing policies targeted at communication and awareness of AD in order to reduce the stigma associated with this form of dementia.

**Data Availability Statement:** The datasets generated and analysed during the current study are available in the Open Science Framework (OFS) repository, https://mfr.osf.io/render?url=https%3A%2F%2Fosf.io%2Fdm6pc%2Fdownload.

**Funding:** The author(s) received no specific funding for this work.

**Competing interests:** The authors have declared that no competing interests exist.

## Introduction

Dementia affects about 50 million people worldwide, of whom 60–70% have Alzheimer-type dementia [1]. Recent research has shown strong evidence of the stigma and discrimination against Alzheimer's disease (AD) [2–4]. Some studies have suggested that the main cognitive attributions leading to stigma in AD are the disease severity and neuropsychiatric symptoms, as well as the lack of aesthetics in people with AD [5, 6]. The most common negative emotions associated with stigma in AD are shame, humiliation and disgust, with compassion considered to be the main positive emotion [7, 8]. These cognitive attributions and negative emotions associated with AD lead to discrimination towards individuals with the disease [9].

Stigma, along with its associated stereotypes, has been identified as one of the barriers to detecting AD and making an early diagnosis [10, 11]. In this regard, a study revealed that general practitioners can overestimate the presence of AD in people with mobility and hearing problems and memory complaints but frequently fail to identify AD in those who lived alone [12]. When a patient is diagnosed with AD, it becomes the main descriptor of who they are because the stigma erases the individual's personality or own personal history, and care systems tend to focus on the dementia rather than on the individual's personal needs [13]. Moreover, the stigmatization of AD also acts as a barrier to the utilization of community services by family members [14–16] or to obtain support from other family members and friends because the disease produces a certain degree of exclusion [17].

The stigma associated with AD and its consequences is reinforced by how the disease is explained or framed [18–20]. This means that negative frames about AD create expectations of discrimination, higher pity and social distance, thereby contributing to stigma [21, 22]. This can be observed, for instance, in behaviours such as prohibiting people with AD from being involved in decision-making processes and often not taking into account their wishes and preferences regarding care [9]. However, framing can also be used as a tool to change attitudes and judgements about AD and thus counteract the stigma and its effects.

### Framing and AD

Framing refers to how a problem is defined [23] or how messages structure an argument in a positive or negative way [24]. For example, messages conveyed about the relationship between a person with AD and their family members may focus on care as a moral value that helps individuals with AD to rediscover their self while keeping them active as long as possible (positive framing). By contrast, messages may emphasize family caregivers' sense of burden, the fact that they receive nothing in return or that they are unable to reconcile care with their life projects (negative framing).

Frames are used to present a topic in such a way that it can be understood by various audiences, that is, different social groups, even those who are not experts in the field. To this end, frames emphasize the central elements shared by society [25]. When interpreting content, individuals are guided by their own mental schemes and cultural processes. Frames are part of the "common ground" of a given culture and therefore do not exist individually [26]. Furthermore, the same event can have different meanings depending on the frame used to define it. Thus, it is understood that one can choose from among several alternatives to define different topics and that the application of a frame results in a specific interpretation of the problem, how it is defined and its causes [27, 28]. The application of a particular frame may lead to a particular interpretation because frames work like a metaphor or an analogy. In this sense, framing offers an interpretation of one topic as equal to another; which sometimes implies simplifying this topic. However, despite the fact that frames constitute an integral part of the culture and everyone has become familiar with frames during the socialization process,

framing is a partially conscious process for both the person who creates a message and the person who receives it [28].

With regard to AD, Van Gorp and Vercruysse conducted a study to determine and analyse the dominant frames for AD [29]. After examining various media content, the authors identified six dominant frames, of which five portray dementia in a negative light.

In Western culture, the dominant and most widely used frame to represent AD is body-mind dualism. This frame posits that human beings are made up of two distinct parts: the body and the soul. When a person has AD, the body remains intact while disease confiscates the soul. According to this view, people with AD lose their identity and humanity by existing only at the material level. Contrary to this frame is the body-mind unity counter-frame. In this frame, the body-mind is conceived as an indivisible whole and therefore the person with AD loses cognitive capacity but not their identity or emotional and sensory capacity. The body-mind unity frame focuses on the capacities that the person with AD retains rather than focusing on what the individual has lost.

Recent studies indicate that the use of positive frames to represent AD, such as body-mind unity, can have a positive influence on attitudes towards AD in both young [18, 30] and adult populations [20, 31]. In this regard, younger people hold more positive attitudes towards people with AD than older people [32]. Furthermore, current research shows that AD stigma differs by age, thus those interventions aimed at reducing AD stigma almost double their effect on younger people compared to the older one [20]. However, research is needed to understand the influence of positive frames on the attitudes of older adults.

## Framing as an instrument of persuasion: The elaboration likelihood model

Attitudes can be affected by the use of various strategies, such as providing arguments [33] or appealing to people's emotions [34]. In the specific context of AD, we know that certain message frames affect attitudes towards the disease [18, 30]. Here, however, we are interested in explaining the mechanism behind these framing effects, the factors involved and the differential effect on groups of people of different ages and under different conditions. To explain the mechanism by which framing affects attitudes, the elaboration likelihood model (ELM) is of particular interest [35, 36]. The ELM proposes that changes in attitudes are the result of psychological processes that depend on the individual's degree of information elaboration at the time the influence is taking place. The degree of elaboration a person engages in should be understood as a continuum that can range from low to high thinking. Factors that influence the individual, such as motivation and the ability to think or process the message (e.g., the presence of distractions), will determine the degree of elaboration. In addition, according to the ELM, attitude change can occur under both low and high thinking conditions but changes produced under high thinking conditions will be enduring and have a greater impact on people's behaviour.

With our study, another interesting aspect of the ELM is the proposal that emotions affect changes in attitude [37]. The ELM proposes that emotions can influence judgements under both low and high elaboration conditions, but in each of these cases the processes by which this influence occurs are different. Under low thinking conditions (when people are not motivated or unable to process information), emotions tend to influence attitudes through low effort processes such as classical conditioning or simple inference. In contrast, under high thinking conditions, the mechanisms responsible for mediating the influence of emotions involve greater effort. Emotions can bias the direction of thoughts that form the person's judgement (thought biases): for example, by retrieving information congruent with their emotional state [38] or by influencing the amount of resources used to process the message.

With regard to the type of emotion that facilitates more elaborate processing, some studies have found that greater message processing is associated with feelings of happiness rather than sadness, but this is not always so. The hedonic contingency theory or view [39] states that individuals in a happy mood want to maintain this mood state and are more sensitive to the happy aspects of the message, whereas they avoid processing messages incongruent with this emotional state. The depth of message processing in a happy state is determined by mood-congruency expectations and the threat the message poses to that positive mood [40]. Nonetheless, we must bear in mind that the age of the person processing the message will also influence the intensity of the emotion, as younger and older people experience different levels of emotional intensity, with younger people experiencing both positive and negative emotions more intensely [41].

The ELM enables the prediction of how a certain frame used to communicate information about AD can produce more enduring changes that have an effect on behaviour and induce high thinking. To achieve this, the message argument must be relevant and consistent with the individual's prior attitudes and experience. In addition, the type of emotion evoked by the frame is expected to influence which arguments the individual will consider and process in greater depth.

The aim of the study was to determine if AD framing influences younger and older people's attitudes towards AD in the same way, as well as the mediating role the emotions induced by the different frames may have in this change of attitude. For this purpose, several AD campaign posters were designed using the body-mind dualism frame and the body-mind unity frame to test the following hypotheses:

- Hypothesis 1: Attitudes towards AD differ according to the age of the participants. Younger people will exhibit more positive attitudes towards AD.

- Hypothesis 2: Messages of the body-mind unity frame will induce favourable attitudes towards AD, in contrast to messages of the body-mind dualism frame, regardless of the audience's age.

- Hypothesis 3: The impact of the message and the emotions reported by the audience will differ depending on the message frame. Specifically, dualism messages are expected to produce high levels of impact, sadness and anger, whereas unity messages will produce high levels of happiness.

- Hypothesis 4: Older participants will experience emotions in a more neutral way than younger participants, who will experience their emotions more intensely.

## Materials and methods

### Participants

One hundred and thirty-six participants of both genders were recruited from the University of Cordoba, Spain. The sample was composed of two age groups: 68 young adults ($M_{age}$ = 18.15, $SD_{age}$ = .95, range = 17–20; 69.12% female) and 68 older adults ($M_{age}$ = 68.03, $SD_{age}$ = 4.94, range = 65–89; 52.94% female). The younger participants were selected from among first-year undergraduate students enrolled in the Bachelor's Degree in Early Childhood Education and Primary Education. The older adults were also university students but at the *Centro Intergeneracional 'Francisco Santisteban'* (University of Cordoba), a higher education programme for older citizens.

The sample size was determined based on the effect size *f* [42] using statistical power analysis program G*Power 3.2.9.7 [43]. The power analysis (*p* = .05, two-tailed, 95% statistical

power) revealed that the total sample should include a minimum of 76 participants to detect a medium effect size ($f = 0.25$) in a mixed-design ANOVA to compare pre-post measures in two groups. We increased the target sample size to take into account possible withdrawals throughout the study.

The study was approved by the research ethics committee of the University of Cordoba (CEIH-21-7). All participants voluntarily agreed to take part in the study. They were informed about all the phases of the study, the tasks to be carried out and that they could withdraw from the research if they so wished. A written 'free and informed consent' form was obtained from all the participants.

## Study variables

All participants were asked to provide their sociodemographic data (age and gender) and to answer two questions: experience with AD using a five-point Likert scale (1 = *I barely know about AD and have never known anyone with AD*; 5 = *I have or have had a family member or close acquaintance with AD and live or have lived with him/her*); and importance attributed to the AD (1 = *Not important*; 5 = *Very important*). The dependent variables were *Attitudes towards AD* and *Self-reported impact and emotions*. On the other hand, the independent variables were *Framing* and *Age group*.

**Attitudes towards Dementia Scale (ADS).**   The ADS [44] assesses attitudes towards dementia and comprises 20 items using a seven-point Likert-type scale from 1 (*Strongly disagree)* to 7 (*Strongly agree*). The ADS score range is 20–140 points, with higher scores indicating more positive attitudes towards dementia. The internal consistency of the scale is good (Cronbach's alpha = .89), as are its goodness-of-fit indices (CFI = .92; GFI = .94; AGFI = .90; PGFI = .91; RMSEA = .05). The ADS was also used in other studies examining the influence of AD framing on attitudes towards AD [18, 30].

**Self-reported impact and emotions.**   To measure the level of impact and basic emotions (happiness, sadness, anger, fear and disgust) experienced by the participants, they were asked six questions designed specifically for this purpose, answered using a five-point Likert scale (1 = *Not at all*; 5 = *Extremely*). Specifically, they were asked: "*What level of impact / happiness / sadness / anger / fear / disgust have you felt after seeing the posters*?" Other research on AD framing has measured the impact and emotions produced by the use of different frames in the same way [18, 30, 31].

**Framing (framing condition).**   This variable was manipulated by means of two conditions that were used in the design of the posters created for the study: dualism frame (messages characteristic of the body-mind dualism frame) and unity frame (messages based on the body-mind unity frame).

**Age group (group condition).**   This variable refers to the participants' age. It is manipulated by selection and presents two conditions: younger adults and older adults.

## Procedure

The present study investigated how message framing in AD influences attitudes towards AD, the impact of these campaigns and the emotions they induce, taking into account the recipients' age. For this purpose, a mixed design was used. Data were collected in each of the two phases of the study: pre-test (Phase 1) and post-test (Phase 2). There was a time interval of 1 week between Phases 1 and 2. The study design and measurement time points are described in Fig 1.

In Phase 1 of the study, all the participants provided information about their sociodemographic data, their experience with AD and the importance of AD for them. They then

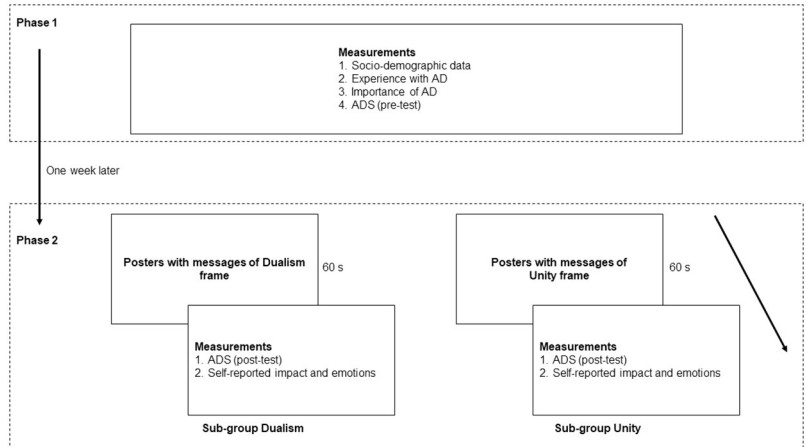

**Fig 1. Study design and measurement times.**

completed the ADS for the first time. A week later, in Phase 2, both the younger and older groups were randomly divided into two sub-groups. Specifically, each participant was assigned a number which was randomly associated with a different subgroup. Each of the sub-groups was shown posters of a campaign with different AD frames: one group viewed posters with messages of the dualism frame and the other group viewed posters with messages of the unity frame. Each group viewed six posters for a period of 1 minute (10 seconds per poster) related to the type of framing to which the group had been assigned. The posters were shown in a silent, isolated room. After viewing the posters, the participants completed the ADS again and responded to questions about the level of impact the campaign had on them and the emotions it evoked.

## Stimuli construction

A series of posters containing both images and text were designed. The posters were presented to the participants as if they formed part of a campaign to raise awareness about AD (Fig 2). Posters were designed for each of the two types of framing condition: dualism and unity. The messages were based on the examples proposed in an inventory of dementia frames and counter-frames [29]. Thus, the only difference between the posters presented to the participants was the message framing. All the messages were written against a black background and the length of each text was short. In addition, to increase the credibility of the message source, the posters contained the logo of the Ministry of Health of the Government of Spain.

The main image on all the posters was a sepia photograph. Photographs of faces of older people were used. The decision to use images of older adults was made based on several studies that point to the positive influence of this type of image in the overall assessment of a campaign because they are considered by the audience to be more impactful, credible and understandable [18, 19]. The same photographs were used for the two conditions. Half of the faces were of men and half of the faces were of women.

## Data analysis

Normality was checked using the Shapiro-Wilk test and by evaluating skewness and kurtosis. To check the homogeneity of the two groups with respect to the sociodemographic variables, both one-way analysis of variance (ANOVA) and the chi-square test ($\chi^2$) were performed,

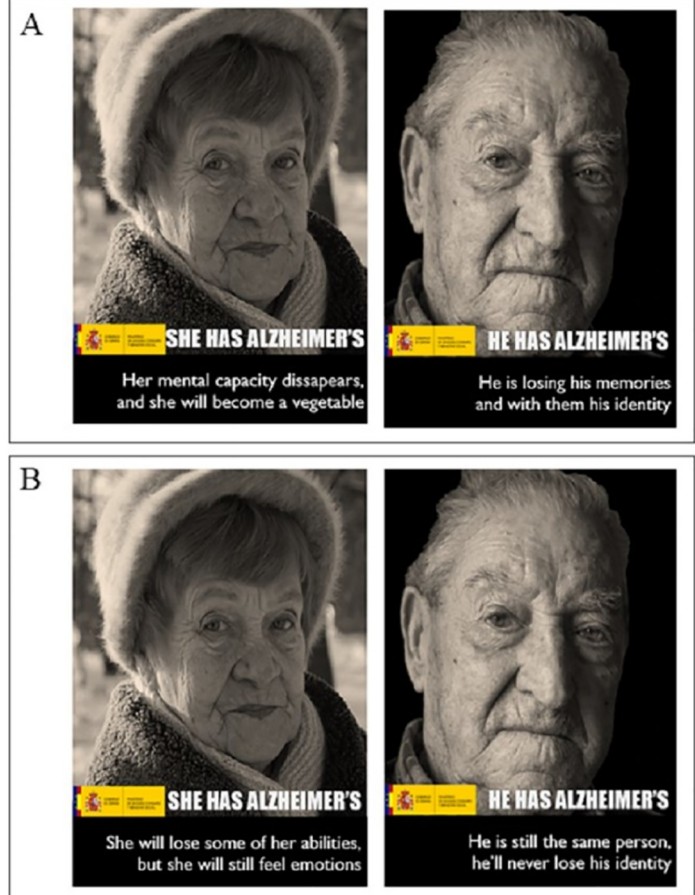

**Fig 2. Examples of posters designed for the study.** All the images used were downloaded from the Pixabay website and were free of copyright under the Creative Commons CC0 license. Panel A: Examples of posters of dualism framing. Panel B: Examples of posters of unity framing. As can be seen, the posters are similar, the only difference is the type of message used. The text of the posters has been translated from Spanish.

depending on the type of variables analysed. *Cramer's V* was the effect size associated with the $\chi^2$ tests. In general terms, $V \leq 0.30$ was considered to be a small effect size, $V = 0.30–0.50$ a medium effect and $V \geq 0.50$ a large effect (Cohen, 1988). In the one-way ANOVA, the effect size was obtained through $\eta^2$, assuming small ($\eta^2 < 0.05$), medium ($\eta^2 = 0.05–0.24$), high ($\eta^2 = 0.25–0.50$) and very high ($\eta^2 > 0.50$) effect sizes [45].

The main effects and interaction estimate of interest on attitudes toward dementia (ADS scores) were analysed using a 2×2×2 mixed-design ANOVA, with 'group' (younger/older) and 'framing' (dualism/unity) as the between-participant variables and 'phase' (Phase 1: ADS pre-test / Phase 2: ADS post-test) as the within-participant variable. When a study condition effect was significant, pairwise contrasts of the estimated marginal mean (EMM) corrected by Bonferroni were used to determine the significance of differences between the levels of the condition in outcome measures. The effect sizes of the main effects and interaction were obtained through $\eta^2$.

A multivariate analysis of variance (MANOVA) was performed with the dependent variables 'impact' and 'emotions' using the 'framing' and 'group' conditions as fixed factors; $\eta^2$ was used to obtain the effect sizes of the main effects and interaction. Finally, Pearson's

correlation coefficient ($r^2$) was used to check whether there was a relationship between attitude change (difference between ADS scores in Phase 1 and Phase 2) and the different emotions.

Data analyses were performed using the Statistical Package for Social Sciences (SPSS) version 21 (IBM®, SPSS Statistics version 21). A *p* value of .05 (two-tailed) was considered to be significant.

## Results

Table 1 displays the results of outcome measures by sociodemographic characteristics and study condition ('group' and 'framing'). Results from ANOVA and chi-square analyses showed that there were no significant differences between the framing data in terms of age, ADS, experience with AD, importance of AD or gender for both younger and older groups (Table 1).

### Effect of framing and group conditions on attitudes towards AD

There was a significant main effect of the framing condition ($F_{(1, 132)}$ = 3.94, $p < .05$, $\eta^2$ = .03) and the phase of study condition ($F_{(1, 132)}$ = 11.88, $p < .001$, $\eta^2$ = .08). The interaction between phase of study and framing (Phase×Framing) was also significant ($F_{(1, 132)}$ = 35.55, $p < .001$, $\eta^2$ = .21). The pairwise contrasts of the EMM revealed significant differences in ADS at post-phase (difference of estimated marginal means [EMMD] = -6.79, $p < .05$, $d$ = -.67, CI = -10.08/-3.51). Therefore, the unity messages caused significantly more positive attitudes toward AD than dualism messages after exposure to the posters. There was also a significant main effect of group condition ($F_{(1, 132)}$ = 17.80, $p < .001$, $\eta^2$ = .12) but the interaction between phase of study and group (Phase×Group) was not significant ($F_{(1, 132)}$ = 0.11, $p > .05$, $\eta^2$ = .001). Thus, the attitudes toward dementia were more positive after exposure to the unity messages in both the younger and older groups (Table 2).

Finally, there were also marginal differences in attitudes due to the triple interaction between study phase, the framing used and participants' age (Phase×Framing×Group): $F_{(1, 132)}$ = 3.69, $p$ = .057, $\eta^2$ = .03. On the one hand, non-significant differences were found in the dualism condition between Phase 1 and Phase 2 on younger adults (EMMD = 0.18, $p >$ .05, $d$ = .02, CI = -2.41/2.77). However, older adults presented lower ADS scores after viewing the dualism campaigns (EMMD = 3.12, $p < .05$, $d$ = .28, CI = 0.53/5.71). On the other hand, in the unity condition, significant differences were found between study phases: both the younger

**Table 1. Sample characteristics, outcomes and study conditions.**

| | Younger Group | | | | | | Older Group | | | | | |
|---|---|---|---|---|---|---|---|---|---|---|---|---|
| | Total sample | Dualism | Unity | $F_{(1, 66)}$ | $p$ | $\eta^2$ | Total sample | Dualism | Unity | $F_{(1, 66)}$ | $p$ | $\eta^2$ |
| | (N = 68) | (n = 34) | (n = 34) | | | | (N = 68) | (n = 34) | (n = 34) | | | |
| | M (SD) | M (SD) | M (SD) | | | | M (SD) | M (SD) | M (SD) | | | |
| Age | 18.15 (0.95) | 18.21 (0.95) | 18.09 (0.97) | .26 | .61 | .00 | 68.03 (4.94) | 67.44 (3.16) | 68.62 (6.23) | .96 | .33 | .01 |
| ADS (20–140)[a] | 102.76 (7.54) | 101.47(6.91) | 104.06 (8.00) | 2.04 | .16 | .03 | 96.84 (10.35) | 99.15 (9.93) | 94.53 (10.38) | 3.51 | .07 | .05 |
| Experience with AD (1–5) | 2.9 (1.12) | 2.71 (1.19) | 3.09 (1.03) | 2.01 | .16 | .03 | 3.03 (1.45) | 3.06 (1.39) | 3 (1.52) | .03 | .87 | .00 |
| Importance of AD (1–5) | 4.21 (0.89) | 4 (0.99) | 4.41 (0.74) | 3.79 | .06 | .05 | 4.56 (0.72) | 4.65 (0.6) | 4.47 (0.83) | 1.02 | .32 | .01 |
| | N (%) | n (%) | n (%) | $\chi^2_{(1)}$ | $p$ | V | N (%) | n (%) | n (%) | $\chi^2_{(1)}$ | $p$ | V |
| **Gender** | | | | | | | | | | | | |
| Female | 47 (69.12) | 25 (73.53) | 22 (64.71) | .56 | .45 | .10 | 36 (52.94) | 20 (58.82) | 16 (47.06) | .94 | .33 | .12 |
| Male | 21 (30.88) | 9 (26.47) | 12 (35.29) | | | | 32 (47.06) | 14 (41.18) | 18 (52.94) | | | |

[a] Scores obtained in phase 1 of the study.

**Table 2. Descriptive statistics on ADS.**

| | | ADS[a] | | |
| | | | Framing | |
| | | | Dualism | Unity |
| Phases study | Group | *M (SD)* | *M (SD)* | *M (SD)* |
|---|---|---|---|---|
| Phase 1 | Younger Adults (*n* = 34) | 102.76 (7.54) | 101.47 (6.91) | 104.06 (8.00) |
| | Older Adults (*n* = 34) | 96.84 (10.35) | 99.15 (9.93) | 94.53 (10.38) |
| | Total (*N* = 68) | 99.80 (9.50) | 100.31 (8.57) | 99.29 (10.38) |
| Phase 2 | Younger Adults (*n* = 34) | 105.24 (8.90) | 101.29 (8.58) | 109.18 (7.44) |
| | Older Adults (*n* = 34) | 98.88 (11.36) | 96.03 (12.17) | 101.74 (9.87) |
| | Total (*N* = 68) | 102.06 (10.66) | 98.66 (10.78) | 105.46 (9.45) |

[a] The score range on ADS is 20 to 140.

participants (EMMD = -5.12, $p < .001$, $d$ = -.66, CI = -7.71/-2.53) and the older adults (EMMD = -7.21, $p < .00$, $d$ = -−.71, CI = -9.80/-4.62) presented higher ADS scores after exposure to the unity posters (Fig 3).

## Self-reported impact and emotions towards the campaigns

The MANOVA result was significant for framing (Pillai's trace = .28, $F_{(6,126)}$ = 8.24, $p < .001$, $\eta^2$ = .28), indicating differences in self-reported impact and emotions. Overall, impact ($F_{(1,135)}$ = 9.95, $p < .01$, $\eta^2$ = .07), sadness ($F_{(1,135)}$ = 16.16, $p < .001$, $\eta^2$ = .11), anger ($F_{(1,135)}$ = 4.95, $p < .05$, $\eta^2$ = .04) and happiness ($F_{(1,135)}$ = 40.39, $p < .001$, $\eta^2$ = .24) were influenced by framing (Table 3). Specifically, the dualism messages were considered more impactful than the unity messages (EMMD = 0.65, $p < .01$, $d$ = .52, CI = 0.24/1.06). With regard to sadness, the dualism messages were considered to be sadder than the unity messages (EMMD = 0.76, $p < .001$, $d$ = .69, CI = 0.39/1.13). Furthermore, dualism messages caused anger to a greater degree than unity messages (EMMD = 0.46, $p < .05$, $d$ = .37, CI = 0.05/0.87). Moreover, the unity-framed messages were considered significantly more happy than the dualism-framed messages (EMMD = 1.23, $p < .001$, $d$ = 1.07, CI = 0.85/1.62), although both groups experienced low-medium levels of happiness after viewing campaigns. The correlations only showed a significant relationship between attitude change and happiness ($r^2$ = .32, $p < .01$).

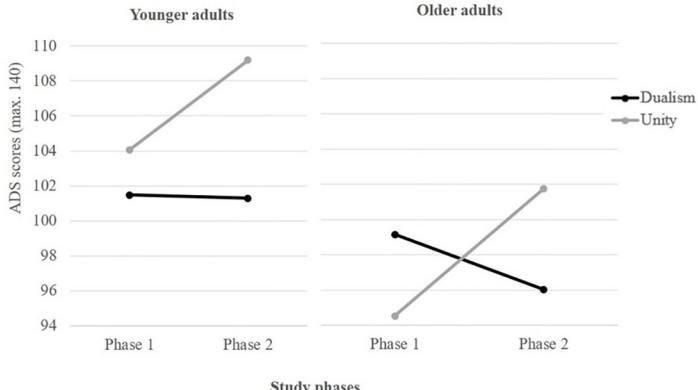

**Fig 3. Effect on ADS of the interaction between framing, audience age, and the study phases controlling for experience with AD and importance of AD.**

**Table 3. Descriptive statistics on impact and emotions self-reported.**

| | | | Framing | |
| --- | --- | --- | --- | --- |
| | | | Dualism | Unity |
| Variable [a] | Group | *M (SD)* | *M (SD)* | *M (SD)* |
| Impact | Younger Adults (*n* = 34) | 3.76 (1.00) | 4.15 (0.76) | 3.38 (1.07) |
| | Older Adults (*n* = 34) | 3.18 (1.42) | 3.44 (1.54) | 2.91 (1.26) |
| | Total (*N* = 68) | 3.47 (1.26) | 3.79 (1.26) | 3.15 (1.19) |
| Happiness | Younger Adults (*n* = 34) | 1.82 (1.22) | 1.06 (0.24) | 2.56 (1.33) |
| | Older Adults (*n* = 34) | 2.19 (1.35) | 1.71 (1.24) | 2.68 (1.27) |
| | Total (*N* = 68) | 2.01 (1.30) | 1.39 (0.95) | 2.62 (1.31) |
| Sadness | Younger Adults (*n* = 34) | 4.10 (1.05) | 4.48 (0.67) | 3.74 (1.21) |
| | Older Adults (*n* = 34) | 3.82 (1.25) | 4.21 (1.04) | 3.44 (1.33) |
| | Total (*N* = 68) | 3.96 (1.16) | 4.34 (0.88) | 3.59 (1.27) |
| Fear | Younger Adults (*n* = 34) | 1.99 (1.24) | 2.48 (1.33) | 1.50 (0.93) |
| | Older Adults (*n* = 34) | 2.09 (1.35) | 2.00 (1.37) | 2.18 (1.34) |
| | Total (*N* = 68) | 2.04 (1.29) | 2.24 (1.36) | 1.84 (1.19) |
| Anger | Younger Adults (*n* = 34) | 2.03 (1.30) | 2.27 (1.33) | 1.79 (1.25) |
| | Older Adults (*n* = 34) | 1.60 (1.12) | 1.82 (1.38) | 1.38 (0.74) |
| | Total (*N* = 68) | 1.81 (1.23) | 2.04 (1.36) | 1.59 (1.04) |
| Disgust | Younger Adults (*n* = 34) | 1.03 (0.17) | 1.03 (0.17) | 1.03 (0.17) |
| | Older Adults (*n* = 34) | 1.07 (0.26) | 1.06 (0.24) | 1.09 (0.29) |
| | Total (*N* = 68) | 1.05 (0.22) | 1.04 (0.21) | 1.06 (0.24) |

[a] The score range on impact, happiness, sadness, anger, fear, and disgust is 1 to 5.

In addition, there were significant MANOVA results for group condition (Pillai's trace = .12, $F_{(6,126)}$ = 2.98, $p <$ .01, $\eta^2$ = .12). Participants' age influenced the level of impact ($F_{(1,135)}$ = 8.23, $p <$ .01, $\eta^2$ = .06) and anger ($F_{(1,135)}$ = 4.33, $p <$ .05, $\eta^2$ = .03) experienced (Table 3). Thus, the younger participants indicated that they experienced more impact (EMMD = 0.59, $p <$ .01, $d$ = .47, CI = 0.18/0.99) and anger (EMMD = 0.43, $p <$ .05, $d$ = .61, CI = 0.02/0.84) than the older participants.

Finally, with respect to interaction between framing and group (Framing×Group), non-significant differences were found (Pillai's trace = .08, $F_{(6,126)}$ = 1.76, $p$ = .11, $\eta^2$ = .08) (Table 3). Nevertheless, only the fear emotion was influenced by this interaction ($F_{(1,135)}$ = 7.25, $p <$ .01, $\eta^2$ = .05). Specifically, the younger adults experienced similar levels of fear after viewing dualism campaigns as the older participants (EMMD = 0.49, $p$ = .12, $d$ = 0.36, CI = -0.12/1.09), whereas the older adults reported higher levels of fear than the younger adults group after exposure to unity messages (EMMD = 0.68, $p <$ .05, $d$ = 0.59, CI = 0.08/1.28). In addition, the older group experienced similar levels of fear under all framing conditions (EMMD = -0.18, $p$ = .56, $d$ = -0.13, CI = -0.78/0.43) (Fig 4).

## Discussion

The influence of framing on attitudes towards AD has been studied in both young [18, 30] and adult populations [20, 31]. To our knowledge, this was the first study in which the attitudes of older adults were analysed and compared to those of younger people by taking into account how the framing of dementia and the emotions influence attitudes. Our analysis highlighted that older adults displayed positive attitudes towards AD, although to a lesser extent than younger people. Nonetheless, both younger and older people's attitudes are more positive after

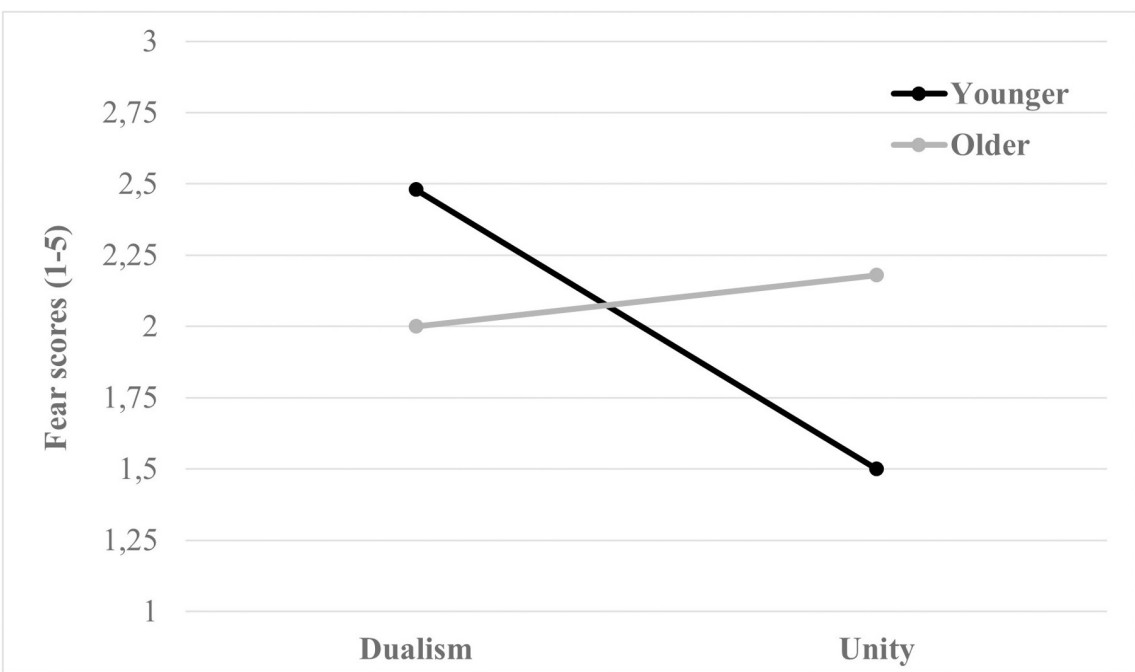

**Fig 4. Effect on self-report fear of the interaction between framing and group.**

exposure to body-mind unity messages. This positive change in attitudes is related to the positive emotion of happiness induced by body-mind unity framing. Therefore, framing impacts on how the audience perceives and responds to a message, and hence it is essentially a fast way to change attitudes and behaviors [46].

### Influence of framing on attitudes towards dementia

With each decade of age, people show increasing compassion and concern for individuals with AD because they believe that those with the disease lack support [3]. This may explain why older people have fewer positive attitudes towards AD than younger people. Understanding the determinants of these concerns about AD in the older population can serve to inform policies and practices to change the social climate affecting people with AD. Older persons are also more concerned about the consequences of AD because they are more likely to have personal experiences with friends or family members with the disease [47] as the risk of AD increases with age. These concerns may reflect their perception of the challenges faced by people with AD.

Our findings provided strong evidence of the influence of framing on attitudes towards AD. Specifically, the unity-framed messages induced a positive change in attitudes, regardless of the audience's age, whereas dualism-framed messages did not yield any change in younger adults and induced a negative change in attitudes towards AD in older adults. This information is key for identifying which characteristics of messages about AD can foster positive attitudes and thus reduce the stigma associated with the disease, therefore, like other authors [20, 29, 30], we believe it is necessary to avoid negative dominant frames and provide positive frames realistic and not idealized (e.g., focusing on the capacities that people with AD retain, such as their identity or emotional and sensory capacity). This should be the case regardless of the age of the population to which the message is addressed.

## Attitude change and ELM: Framing and emotions

In our case, we knew that the participants were aware of AD, considered it an important issue and had positive attitudes towards the disease prior to conducting the study, as they had obtained high ADS scores prior to the exposure to the posters. In addition, the context in which the message was presented was specifically designed to facilitate information processing. The participants were given enough time to process the message, the message was presented in a very clear manner and we motivated and instructed them throughout the process. Thus, we assume that all the conditions were met to ensure high thinking during the message processing.

Once the conditions were established to create a situation of influence or persuasion, we introduced two types of frames: one that was consistent with the stigma associated with AD (dualism) and another that was not consistent with the stigma (unity). The dualism-framed messages caused a greater impact and induced more negative emotions (sadness and anger) in both the younger and older participants but with anger being more intense in the younger participants. In contrast, the unity frame had less impact but evoked higher positive emotions (e.g., happiness) scores, even though these were not intense. Even so, this slight increase was enough to achieve a change in the attitude towards AD.

It is important to highlight that the emotions reported by the participants were induced by the message itself. This aspect constitutes a differential factor with respect to other studies in which emotions are elicited by means of activities or tasks other than the arguments themselves. The fact that the groups shown the different frames reported different emotions indicates that the messages were processed in great depth, as this would not have occurred otherwise [34]. Indeed, in our framing, there is no possibility of a simple association or a simple inference effect between the emotion and the object of the attitude, as processing the argument is a necessary condition for experiencing one or another emotion. However, although both the dualism-framed and unity-framed messages were processed in depth, only those that evoked positive emotions improved attitudes towards AD. In these situations, emotions impact on attitudes in line with their valence. That is, if the attitude object is associated with a positive emotional state or feeling (e.g., happiness), that object will be more appreciated than if it would be associated with a negative emotional state or feeling (e.g., fear) [35, 48].

While it is true that AD health campaigns, especially those associated with charitable fundraising, have often appealed to emotions and relied on fear-based approaches, this method may actually be contrary to the goals of campaigns aimed at mitigating stigma [30, 31]. Our findings show that the older adults experience similar levels of fear after exposure to AD campaigns, regardless of the frames used. Thus, attitude change is not produced by appealing to fear. Such campaigns can also undermine support and empathy for people living with and caring for those with AD. Our conclusions are along these lines, as they indicate that impactful messages do not affect people's attitudes towards AD but, rather, lead to feelings of sadness or anger. However, both younger and older people do experience a change in attitude when the message includes less impactful arguments that elicit positive, happy emotions that make them feel good.

Both younger and older people were influenced by unity messages and changed their attitudes towards AD, thus the frames worked similarly regardless of the age of the audience. Specifically, the body-mind unity frame helped reduce stigma more than negative frames such as the mind-body dualism. Likewise, these frames evoked different emotions that contributed to the change in the attitudes. However, emotions such as anger or fear were not reported in the same way by younger and older people. Therefore, more research is needed in this regard to

support more firmly an age-targeting approach in advocacy and campaigns, as other authors have done [20].

### Limitations, future directions and strengths

This study has some limitations. First, it was conducted in a single setting and participants' experiences and motivations may vary depending on the context. In addition, the research provides insight into the most widely used frames in Western culture, so it is not possible to generalize the results to other countries where there may be differences in how the messages are interpreted or in the framing used. It would have been interesting also to know the participants' degree of responsibility for those with the disease and their care. We must continue to expand our knowledge about how AD is represented. The frames used and their influence on the population need to be studied further. In addition, it would be very interesting to carry out the same study with a sample that includes caregivers, oldest-old (aged years and over), people with AD and people from other cultures.

The results of this study can serve to inform both formal and informal institutional communication campaigns. Communication about AD using positive frames such as unity will help to improve attitudes towards dementia in the general population. Reframing of AD is needed through the use of positive frameworks in different media, health campaigns and the available health resources.

### Acknowledgments

BF-C was the recipient of a Senior Distinguished Researcher Position (Beatriz Galindo Programme) in the Department of Psychology at the Universidad de Córdoba (ref. BEAGAL18/00006).

### Author Contributions

**Conceptualization:** Fátima Cuadrado, Adoración Antolí.

**Data curation:** Fátima Cuadrado, Adoración Antolí.

**Formal analysis:** Fátima Cuadrado, Bernardino Fernández-Calvo.

**Investigation:** Fátima Cuadrado, Adoración Antolí.

**Methodology:** Fátima Cuadrado, Bernardino Fernández-Calvo.

**Supervision:** Adoración Antolí, Bernardino Fernández-Calvo.

**Writing – original draft:** Fátima Cuadrado, Adoración Antolí.

**Writing – review & editing:** Fátima Cuadrado, Adoración Antolí, Bernardino Fernández-Calvo.

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
