## [Decision Letter · Decision Letter 0]

6 Apr 2022

PONE-D-22-09110What does Alzheimer’s disease mean for younger and older people? An approach to the effect of framingPLOS ONE

Dear Dr. Cuadrado,

Thank you for submitting your manuscript to PLOS ONE. After careful consideration, we feel that it has merit but does not fully meet PLOS ONE’s publication criteria as it currently stands. Therefore, we invite you to submit a revised version of the manuscript that addresses the points raised during the review process.

We look forward to receiving your revised manuscript.

Kind regards,

Perla Werner

Academic Editor

PLOS ONE

Journal Requirements:

2. We note that Figure 2 in your submission contain copyrighted images. All PLOS content is published under the Creative Commons Attribution License (CC BY 4.0), which means that the manuscript, images, and Supporting Information files will be freely available online, and any third party is permitted to access, download, copy, distribute, and use these materials in any way, even commercially, with proper attribution. For more information, see our copyright guidelines: http://journals.plos.org/plosone/s/licenses-and-copyright.

3. Please ensure that you include a title page within your main document. You should list all authors and all affiliations as per our author instructions and clearly indicate the corresponding author.

Reviewers' comments:

Reviewer's Responses to Questions

**Comments to the Author**

1. Is the manuscript technically sound, and do the data support the conclusions?

Reviewer #1: Yes

Reviewer #2: Yes

2. Has the statistical analysis been performed appropriately and rigorously? 

Reviewer #1: Yes

Reviewer #2: Yes

3. Have the authors made all data underlying the findings in their manuscript fully available?

Reviewer #1: Yes

Reviewer #2: Yes

4. Is the manuscript presented in an intelligible fashion and written in standard English?

Reviewer #1: Yes

Reviewer #2: Yes

5. Review Comments to the Author

Reviewer #1: Thanks for this opportunity to review an interesting paper. Please find my comments below.

1. The current title (What does Alzheimer’s disease mean for younger and older people?) is a bit misleading. I would suggest authors to modify.

2. On page 3, “it is understood that one can choose from among several alternatives to define different topics and that the application of a frame results in a specific interpretation of the problem, how it is defined and its causes”.

Would it be possible to explain a bit more on the application of a frame may lead to a particular interpretation?

3. On page 8, “to measure the level of impact and basic emotions (happiness, sadness, anger, fear and disgust) experienced by the participants, they were asked six questions designed specifically for this purpose, answered using a five-point Likert scale”. Would it be possible to add the question asking impact?

4. Was data collected before the Covid-19 pandemic? If data was collected during pandemic, it would be appreciated that authors could consider/briefly discuss the Covid-related impacts on methodology.

5. On page 19, “it would be very interesting to carry out the same study with a sample that includes caregivers, people with AD and people from other cultures.” Older participants in the study are the young-old. It would also be interesting to conduct a study with sample of the oldest-old (aged 85 years and over).

6. As found in this study, there are age differences in attitudes towards AD, experiencing emotions and etc. Would it be possible to discuss the implications of age differences in relation to choosing frames and messages to reduce stigma? Would the findings support an age-targeting approach in advocacy and campaigns?

Reviewer #2: Manuscript Number: PONE-D-22-09110

Manuscript Title: What does Alzheimer’s disease mean for younger and older people? An approach to the effect of framing

Thank you for the opportunity to review the article titled “What does Alzheimer’s disease mean for younger and older people? An approach to the effect of framing”. The paper is very well written on an important topic. The article reports the results of a study that aimed to determine how AD framing influences attitudes towards AD and whether this influence differs between younger and older people. The study also seems well executed.

I have some comments that might help strengthen the paper. I list conceptual and stylistic issues first then, under minor concerns, list a few that are more technical in nature. Within each section, my comments are listed in no particular order.

Conceptual and stylistic issues:

The introduction overall is strong. However, the characterization of Alzheimer’s would benefit from further development. For example, page 9 “and problematic behaviours of this dementia,” sounds more like behavioral variants of dementia rather than Alzheimer’s dementia.

In addition, also on Page 9 “In this regard, general practitioners overestimate the presence of AD in people with mobility and hearing problems and memory complaints but frequently fail to identify AD in those who lived alone [12].” Perhaps softening and more closely specifying this sentence – “a prior study showed general practitioners can overestimate the presence … “ There are other confounders that are not being acknowledged such as availability of a clinical informant who is also a caregiver that might also explain these findings.

Page 10, “Frames are used to present a topic in such a way that it can be understood by various audiences.” � Do the authors mean: “Frames are used to present a topic in such a way that it can be understood by an audience.”

The concept of happiness warrants explicit discussion, given the context involves a debilitating, chronic neurodegenerative disease.

Extant research shows that AD stigma differs by age group (Stites et al 2018). The current study is not ideally positioned to offer evidence of population age differences. However, the current study is well positioned to build upon prior work to answer the question: What are the implications of age differences on interventions that are aimed at reducing AD stigma? The study shows that the intervention had almost double the effect on the younger group as it did the older group (Table 2).

Stites, S. D., Johnson, R., Harkins, K., Sankar, P., Xie, D., & Karlawish, J. (2018). Identifiable Characteristics and Potentially Malleable Beliefs Predict Stigmatizing Attributions Toward Persons With Alzheimer’s Disease Dementia: Results of a Survey of the U.S. General Public. Health Communication, 33(3), 264–273. https://doi.org/10.1080/10410236.2016.1255847

Page 14, please include rationale for appropriateness of the effect size of 0.25. It may also be appropriate to revisit this issue in the discussion as it relates to study limitations and interpretation of the meaningfulness of the results.

Page 17, “responded to questions about the level of impact the campaign had on them” This aspect of the study is unclear. Can you give examples of these questions?

Page 23 – “This information is key for identifying which characteristics of messages about AD can foster positive attitudes and thus reduce the stigma associated with the disease, therefore, like other authors [20, 29, 30], we believe it is necessary to avoid negative dominant frames and instead employ those that humanize people with AD. This should be the case regardless of the age of the population to which the message is addressed.” The conflation of ideas of “positive attitudes” and “humanizing people” may need to be more closely parsed as the notion of ‘positivity’ in the context of a serious neurodegenerative disease could be problematic and even damaging, and, simultaneously, humanizing individuals is not synonymous with ‘positivity’. I point out this issue here but the it applies at multiple points in the paper.

Minor comments:

Abstract “This findings suggest” � These?

Quantification and statistical test supporting the conclusions would be useful. In addition, paper is framed around examining differences between older and younger respondents but abstract doesn’t report whether the frames examined in this study differed for the two age groups.

In the methods, please add age ranges to help define the study groups of younger adults and older adults.

Cell sizes in the tables would be useful.

Page 24 – “had positive attitudes towards the disease” examples here would help.

Page 17, younger and older groups were randomly divided into two sub-groups. � how was the randomization carried out?

6. PLOS authors have the option to publish the peer review history of their article (what does this mean?). If published, this will include your full peer review and any attached files.

Reviewer #1: No

Reviewer #2: No

---

## [Author Response · Author response to Decision Letter 0]

10 Jun 2022

Firstly, I would like to thank you for reviewing our work and giving us the opportunity to review our paper (Manuscript ID: PONE-D-22-09110).

We have written a point by point cover letter indicating the changes we have made in the manuscript and the responses to each of journal requirements and the comments of the reviewers.

1. We have ensured that our manuscript meets PLOS ONE’s style requirements (e.g. figures and tablets format).

2. With respect to Figure 2, all the images used were downloaded from the Pixabay website and were free of copyright under the Creative Commons CC0 license, so these images can be used and published freely. We have provided this information in Fig 2 legend.

3. We have ensured that our title page meets all requirements.

4. We have included citations 42 and 48, which have been included in the reference list. We have marked them in red so you can see it.

Response to reviewer 1:

1. As suggested, we have modified the title of manuscript (before: What does Alzheimer’s disease mean for younger and older people? An approach to the effect of framing; now: The effect of framing on attitudes towards Alzheimer’s disease. A comparative study between younger and older people).

2. With regard to your comment about the frames interpretation, we have explained more on the application of a frame may lead a particular interpretation. (Page 4-5, lines 76-82).

3. As suggested, we have added the question asking impact and emotions experienced by the participants. (Page 10, lines 212-213).

4. The data was not collected during the pandemic, so we cannot discuss the Covid-related impacts on methodology, since there was no restriction or rule related to the pandemic that prevented the normal development of data collection.

5. We agree with you. Older participants in this study are the young-old, so we have added the oldest-old like an interesting sample that includes in future studies. (Page 23, line 478).

6. As suggested, we have discussed the implications of age differences in relation to choosing messages framed to reduce stigma. (Page 22-23, lines 459-466).

Response to reviewer 2:

Conceptual and stylistic issues:

1. Regarding your comment, we have been more thorough with the characterization of Alzheimer’s disease. For example, we have explained that Alzheimer’s behavioural and psychological disorders (neuropsychiatric symptoms) have been associated with stigma towards AD (Page 3, lines 31-34).

2. We have softened the sentence as you suggested (Page 3, lines 40-41).

3. Regarding your doubt about if we men “an audience” instead of “various audiences”, we have specified what we mean which “various audiences” (Page 4, lines 68-69).

4. We understand the incongruence that you indicate about happiness, but we need answer about all basic emotions (happiness, sadness, fear, anger, and disgust). In this context (chronic neurodegenerative disease), we know that people don’t answer with high levels of happiness, and we don’t intend it. On the other hand, some body-mind unity frame’s affirmations (e.g. she will lose some of her abilities, but she will still feel emotions) can provoke feeling of consolation or tenderness; in short, emotions congruent with happiness. In this way, we have explicated it. Moreover in “Results” section, we have indicated that both groups experienced low-medium levels of happiness (Page 18, lines 355-356).

5. We have included the information provided by you about Stites et al. (2018). (Page 5, lines 99-102).

6. We have specified more about the effect size of 0.25 and we have referenced it. (Page 9, lines 179-180 and lines 182-183).

7. As suggested, we have included an example of questions about level of impact the campaign had on participants. (Page 10, lines 212-213).

8. With regard your comment about “positivity” and “positive messages”, we have clarified what we mean and eliminated confusing concepts. (Throughout the “Discussion” section, e.g. page 20, lines 410-412).

Minor comments:

1. With regard your comments about abstract: (1) we have corrected the abstract’s error (‘these’ rather ‘this’) (page 1, line 21), (2) we have added the statistical test (Page 1, line 13 and 15-16), and (3) we have reported more information about frames examined for the two age groups (Page 1. Lines 9-12).

2. Age ranges are added. We have underlined them in yellow so you can see them. (Page 8, lines 173 and 174).

3. Regarding your comment about cell sizes in the tables, we have included sample size.

4. With respect your comment about illustrate with examples, we have clarified why we knew that the participant had positive attitudes towards disease. (Page 21, lines 418).

5. With regard your question about how the groups were randomly divided, we have explained how the randomization was carried out (Page 11, lines 238-240).

We hope that these changes have improved the manuscript and you will consider it for publication in PLOS ONE.

Yours sincerely,

---

## [Editor Report · Decision Letter 1]

22 Jun 2022

The effect of framing on attitudes towards Alzheimer’s disease. A comparative study between younger and older adults

PONE-D-22-09110R1

Dear Dr. Cuadrado,

We’re pleased to inform you that your manuscript has been judged scientifically suitable for publication and will be formally accepted for publication once it meets all outstanding technical requirements.

Kind regards,

Perla Werner

Academic Editor

PLOS ONE
---

## [Editor Report · Acceptance letter]

27 Jun 2022

PONE-D-22-09110R1 

The effect of framing on attitudes towards Alzheimer’s disease. A comparative study between younger and older adults 

Dear Dr. Cuadrado:

I'm pleased to inform you that your manuscript has been deemed suitable for publication in PLOS ONE. Congratulations! Your manuscript is now with our production department. 

Kind regards, 

on behalf of

Professor Perla Werner 

Academic Editor

PLOS ONE